# Enhancing Solid-Phase Extraction of Tamoxifen and Its Metabolites from Human Plasma Using MOF-Integrated Polyacrylonitrile Composites: A Study on CuBTC and ZIF-8 Efficacy

**DOI:** 10.3390/nano14010073

**Published:** 2023-12-26

**Authors:** Domingo R. Flores-Hernandez, Héctor Manuel Leija Gutiérrez, Jose A. Hernandez-Hernandez, José Antonio Sánchez-Fernández, Jaime Bonilla-Rios

**Affiliations:** 1Tecnologico de Monterrey, Escuela de Ingeniería y Ciencias, Ave. Eugenio Garza Sada 2501, Monterrey 64849, Mexico; a00819488@tec.mx; 2Universidad Autónoma de Nuevo Leon, CICFM-FCFM. Av. Universidad S/N, Ciudad Universitaria, San Nicolas de los Garza 66451, Mexico; hector.leijagt@uanl.edu.mx; 3Tecnologico de Monterrey, Escuela de Medicina y Ciencias de la Salud, Monterrey 64710, Mexico; j.a.hernandez@tec.mx; 4Procesos de Polimerización, Centro de Investigación en Química Aplicada, Blvd. Enrique Reyna No. 140, Saltillo 25294, Mexico

**Keywords:** metal organic frameworks, MOF, nanocomposite, tamoxifen, nanofibers, electrospinning, CuBTC, ZIF-8, solid-phase extraction

## Abstract

This study investigates electrospun fibers of metal-organic frameworks (MOFs), particularly CuBTC and ZIF-8, in polyacrylonitrile (PAN) for the solid-phase extraction (SPE) of Tamoxifen (TAM) and its metabolites (NDTAM, ENDO, and 4OHT) from human blood plasma. The focus is on the isolation, pre-concentration, and extraction of the analytes, aiming to provide a more accessible and affordable breast cancer patient-monitoring technology. The unique physicochemical properties of MOFs, such as high porosity and surface area, combined with PAN’s stability and low density, are leveraged to improve SPE efficiency. The study meticulously examines the interactions of these MOFs with the analytes under various conditions, including elution solvents and protein precipitators. Results reveal that ZIF-8/PAN composites outperform CuBTC/PAN and PAN alone, especially when methanol is used as the protein precipitator. This superior performance is attributed to the physicochemical compatibility between the analytes’ properties, like solubility and polarity, and the MOFs’ structural features, including pore flexibility, active site availability, surface polarity, and surface area. The findings underscore MOFs’ potential in SPE applications and provide valuable insights into the selectivity and sensitivity of different MOFs towards specific analytes, advancing more efficient targeted extraction methods in biomedical analysis.

## 1. Introduction

Cancer is the second major cause of death globally, with around 10 million deaths per year. In this regard, breast cancer is the most common cancer in women and one of their main causes of death [1]. More than 90% of breast cancers are not metastatic when diagnosed. Endocrine therapy (oral antiestrogen administration) is the primary systemic therapy used in these cases. Tamoxifen (TAM) is used as the mainstay hormonal treatment for breast cancer [2,3,4]. Initially, TAM is predominantly metabolized into N-desmethyltamoxifen (NDTAM) and, to a lesser extent, into 4-hydroxytamoxifen (4OHT). These metabolites are further metabolized into Endoxifen (ENDO). The antiestrogenic activity of TAM is primarily due to the presence of the metabolites 4OHT and ENDO. Recent studies have suggested that a positive clinical outcome of TAM hormonal therapy depends on achieving a threshold ENDO concentration, which broadly varies among patients [5,6,7,8,9]. Moreover, TAM can present an estrogen agonist activity in a tissue-specific manner with coactivator proteins, increasing the risk of developing endometrial cancer [10,11]. Therefore, the drug monitoring of TAM and its major metabolites could be a clinically useful tool to achieve an optimal pharmacological response by properly adjusting the TAM doses in every individual [12].

The screening of anti-estrogenic agents predominantly relies on high-pressure liquid chromatography-tandem mass spectrometry (HPLC-MS/MS). This method, while accurate, is costly, time-intensive, demands specialized skills, and is limited in processing large volumes of samples daily [13]. In line with global efforts to ensure equitable healthcare access for all [14], there is a growing need for diagnostic technologies that are both affordable and practical, even in remote areas. To this end, the World Health Organization (WHO) established the ASSURED criteria—Affordable, Sensitive, Specific, User-friendly, Rapid and Robust, Equipment-free, and Deliverable to end-users—as a benchmark for evaluating point-of-care (POC) devices [15]. Recently, paper-based analytical devices (PADs), well-established in the realm of diagnostics, have gained renewed attention as biosensors, as advances in materials science allow for the introduction of novel materials with high-surface areas, customizable pore structures, and improved colorimetric or electrochemical responses. These new functionalities, combined with the PADs’ capillary action, cost-effectiveness, biocompatibility, and ease of customization, enable them to perform complex tasks within a single device and align with the ASSURED criteria [16,17]. In this context, we explore electrospun materials, which possess the various intrinsic and promising properties of PADs as they share the same fibrous polymeric structure. Electrospinning is a process that creates fine fibers from a polymer solution or melt using a high-voltage electric field, producing nanometer to micrometer-scale fibers. The electrospinning process allows the direct addition of functional materials to the polymeric solutions to produce fibrous mats with specific properties [18]. The remarkable attributes of these fibrous materials and their potential in POC applications have sparked interest in exploring the fundamentals of robust analytical platforms. This knowledge could be pivotal in developing a high-performance, nanofibrous-based, solid-phase extractor (SPE), potentially integrating it into a biosensing platform.

SPE is a widely employed analytical technique designed for separating and concentrating specific compounds from complex matrices. The fundamental SPE process involves introducing a solution containing the analytes into a solid phase, typically a cartridge filled with porous sorbent material that retains the target analytes. This is followed by removing unwanted components through washing and then extracting the desired analytes into a collection tube using different solvents [19]. SPE’s versatility and efficiency in sample preparation make it a valuable tool in various fields, including clinical, environmental, pharmaceutical, biochemistry, and food safety analysis. Its ability to effectively purify and concentrate analytes while removing interfering substances enhances the sensitivity and accuracy of subsequent analytical techniques, such as chromatography and mass spectrometry [20].

The primary objective of this research is to harness the unique chemical and physical attributes of metal-organic frameworks (MOFs). MOFs are an emerging class of crystalline materials composed of transition metal ions coordinated with organic linkers with a periodic nano-scaled structure with ultrahigh porosity (up to 90% free volume; pore window ranging from 5 to 25 Å) and exceptionally large surface areas [21,22]. By integrating these properties with advanced nano-fabrication methodologies, such as electrospinning, the intention is to devise a paper-based biosensing platform envisioned to facilitate the isolation, pre-concentration, and extraction of TAM, NDTAM, 4OHT, and ENDO from human blood plasma (Figure 1a width of the arrows in this image indicates the extent of metabolism from one compound to another) [23,24]. Furthermore, the developed composite is designed to align with the ASSURED criteria the World Health Organization sets forth, offering a viable alternative to costly and complex chromatographic techniques.

In this study, we fabricated two distinct composites by electrospinning polyacrylonitrile (PAN) solutions, chosen for their ideal thermal, mechanical, and chemical stability and low density, combined with CuBTC and ZIF-8. CuBTC, a metal-organic framework (MOF), features a three-dimensional structure with open metal sites, ample pore volume, and surface area (Figure 1b); yellow and orange spheres represent pores in CuBTC) [25]. Its selection is based on its high thermal stability, reversible adsorption-desorption without structural degradation, and 64 adsorption sites per unit cell for molecular access [26]. ZIF-8, another MOF, comprises zinc and 2-methylimidazole ligands, mirroring the sodalite structure of inorganic zeolites. It is characterized by uniform porosity, superior chemical and thermal stability, effective dispersion, flexible structure, and a high loading capacity [21,27]. The composite SPE materials were used to explore their interactions with TAM, NDTAM, ENDO, and 4OHT, evaluating their retention and elution efficiency, as well as sensitivity and selectivity. These interactions and performances were rigorously tested under both ideal conditions, where the drugs were dissolved in pure water-methanol blends and human blood plasma. This comprehensive analysis compared the sample pretreatment using ZnSO_4_, acetonitrile, and methanol for protein precipitation, methanol, and blends of acetonitrile and water for the elution process. The overall process is depicted in Figure 2.

## 2. Materials and Methods

### 2.1. Materials

Basolite^®^ C300 (CuBTC), Basolite^®^ Z1200 (ZIF-8), polyacrylonitrile (PAN, average Mw = 150,000), *N,N*-Dimethyl formamide (DMF), and (*Z*)-1-(*p*-Dimethylaminoethoxyphenyl)-1,2-diphenyl-1-butene (Tamoxifen citrate, TAM) were purchased from Sigma-Adrich (Saint Louis, MI, USA). (*Z*)-2-(*p*-(1,2-Diphenyl-1-butenyl)phenoxy)-*N*-methyl ethylamine (*N*-desmethyl tamoxifen, NDTAM), (*E/Z*)-4-[1-[4-[2-(Methylamino)ethoxy]phenyl]-2-phenyl-1-buten-1-yl]-phenol hydrochloride hydrate (1:1 *E/Z* mixture) Endoxifen, ENDO), (*Z*)-4-(1-[4-(Dimethylaminoethoxy)phenyl]-2-phenyl-1-butenyl)phenol (4-Hydroxytamoxifen, 4OHT), and 4-[1-[4-[2-(Methylamino)ethoxy]phenyl]-2-phenyl-1-butenyl-d5]phenol *(E/Z* Endoxifen-d5) were purchased from United States Biological (Salem, MA, USA). 2-[4-[(1*Z*)-1,2-diphenyl-1-buten-1-yl-3,3,4,4,4-d5]phenoxy]-*N*,*N*-dimethyl-ethanolamine (Tamoxifen-d5) and 4-[1-[4-[2-(dimethylamino)ethoxy]phenyl]-2-phenyl-1-buten-1-yl-3,3,4,4,4-d5]-phenol (4-Hydroxytamoxifen-d5) were obtained from Cayman Chemical Company (Ann Arbor, MI, USA). As this research does not involve clinical trials, the biological samples were altruistically donated following our Institutional Research and Ethics Board recommendations.

### 2.2. Instrumentation

The electrospinning setup comprises a power supply (CZE1000PN30 Rack Mounted, Spellman High Voltage Electronics Corporation, Hauppauge, NY, USA), a syringe pump (NE-1600 Six Channel Programmable Syringe Pump, New Era Pump Systems, Farmingdale, NY, USA), and a five-cc syringe adapted with a chamfered syringe tip gauge 18 (Nordson EFD, Westlake, OH, USA).

The morphology of the fibers was studied using a scanning electron microscope (SEM) by Zeiss EVO MA25 with an accelerating voltage of 20 kV after sputtering the sample with a 5 nm Au layer. Image processing software ImageJ (1.54h) by NIH was used to measure the fiber diameters from the SEM micrographs. The fiber diameters were measured at 60 points within the SEM image field for each sample.

Energy-dispersive X-ray spectroscopy (EDS) was used for the elemental composition analysis of the composites. For this purpose, an EDS detector is coupled to the Zeiss EVO MA25 (XFlash 6|10, Bruker, MA, USA).

Fourier transform infrared spectroscopy (FTIR) using a Perkin Elmer Universal (Hopkinton, MA, USA) ATR sampling accessory frontier spectrometer was employed.

The UPLC-MS/MS analysis was conducted using an Acquity UPLC H-Class System coupled to a Xevo TQD tandem mass spectrometer (Waters, MA, USA). The ESI was operated in positive ionization mode. Multiple ion monitoring chromatograms were acquired using MassLynx Mass Spectrometry Software 4.1 (Waters).

### 2.3. Preparation of MOF/PAN Electrospun Composites

The method for producing the electrospun mats was adapted from the work of Chaohai et al., where the weak mechanical properties of pure MOFs were overcome by combining them with a polymeric PAN matrix via electrospinning [28]. Pure PAN fibers were fabricated as a control using 0.525 g (10 wt.%) of PAN in 5 mL of DMF stirred at 700 rpm and 85 °C for 5 h. The polymeric solutions containing the MOF were prepared using the weight of pure PAN solutions as a reference. For MOF/PAN solutions, 0.75 g (12.5 wt.% of the final solution weight) of the specific MOF, CuBTC, or ZIF-8 was added to 5 mL of DMF. This solution was sonicated for 25 min until the MOFs were well dispersed into the solvent. Then, 0.525 g of PAN was added to the solution and stirred at 700 rpm and 85 °C for 5 h to obtain the electrospinning precursor. The electrospinning process was carried out for the CuBTC/PAN solution with a positive voltage of 16 kV with a collecting distance of 17 cm and a polymer flow rate of 0.35 mL/min. For the ZIF-8/PAN solution, a positive voltage of 15.2 kV with a collecting distance of 18 cm and a polymer flow rate of 0.27 mL/h was used. For the pure PAN solution, a positive voltage of 15.5 kV with a collecting distance of 17 cm and a polymer flow rate of 0.17 mL/h was used. In all cases, the room temperature was 24 °C, and the humidity was controlled around 36%. The electrospinning process was not achievable for humidity values greater than 41%.

## 3. Results and Discussion

### 3.1. SEM Characterization

Figure 3a,b showcases the micrographs of the pristine MOFs. The CuBTC particles predominantly fall within the microscale domain, with certain crystals surpassing 100 µm in length. While the characteristic octahedral shape of CuBTC is observable in some of the crystals, it is not universally well defined. CuBTC morphology aligns with previously published SEM images of this material [26,29]. Conversely, ZIF-8 nanoparticles exhibit a consistent size of approximately 200 nm and a distinct cubic shape. These SEM images of ZIF-8 are in concordance with existing studies in the scientific literature [21,28].

Moreover, Figure 3c–e shows the morphology of the CuBTC/PAN, ZIF-8/PAN, and PAN electrospun fibers, respectively. The CuBTC/PAN fibers display a smooth surface, while the ZIF-8/PAN composite exhibits rugosity; this might be attributed to the difference in MOF crystal size. The microcrystals from the CuBTC cannot be completely embedded into the PAN fiber and result in smooth fibers interconnecting CuBTC particles, while the nanocrystals of ZIF-8 can be more evenly distributed inside the fibers. Nevertheless, neither composite shows good dispersity along all the fiber lengths. Standard PAN fibers exhibit a smooth surface with small furrows that are widely reported in the literature [30,31].

Additionally, Figure 3f details the distribution of the diameters of the composites and PAN electrospun fibers. Fifty random measurements were taken using different SEM images of each sample, using the software ImageJ (1.54h) to obtain the descriptive statistics. As observed, CuBTC/PAN fibers have the largest diameters, averaging around 1050 nm. Moreover, it shows the widest distribution of diameters with a large standard deviation. ZIF-8/PAN fibers have, in general, smaller diameters than those from the CuBTC composite, with a mean value of around 650 nm and a narrower diameter distribution. In the case of pure PAN fibers, the diameters are the smallest of all the samples, with an average of around 450 nm and the narrowest diameter distribution. Crystals of different sizes might explain the difference in diameters between CuBTC and the other MOFs. The PAN traps the large particles from CuBTC during the electrospinning process. The capillary action of the polymer solution rich in DMF at the so-called Taylor cone promotes the total entrapment of the particle, leading to significant accumulations of polymer around them. Even when the fibers are stretched due to the bending instabilities on their path to the collector, these large quantities of polymer end up with big diameters. Moreover, the wide distribution of crystal sizes of CuBTC supports this hypothesis since this composite also has the widest diameter distribution. On the other hand, the smaller and more consistent size of ZIF-8 crystals results in smaller diameters and narrower diameter distributions.

### 3.2. Surface Energy Characterization

Figure 4a–c depicts the surface energy characteristics for PAN, CuBTC/PAN, and ZIF-8/PAN filters, respectively, utilizing the Owens, Wendt, Rabel, and Kaelble (OWRK) method. We employed water and methanol to determine the filters’ surface energy polar and non-polar components, providing insights into their interaction potential with different analytes.

The polyacrylonitrile (PAN) filter exhibits a surface energy of approximately 50 mN/m. This energy is entirely assigned to the dispersive term of the OWRK model, reflecting the non-polar nature of PAN’s molecular structure. This suggests that the PAN filter is predisposed to interact predominantly with non-polar substances, given its lack of a significant polar component in surface energy.

In contrast, the CuBTC/PAN composite filter demonstrates a lower surface energy than pure PAN, around 30 mN/m. While most of this energy remains dispersive, similar to PAN, a third of the surface energy is attributed to the polar component. This small increase in polar character indicates an enhanced capacity for interacting with polar analytes, although it is still primarily in a non-polar interaction domain.

Interestingly, the ZIF-8/PAN composite shows a surface energy parallel to PAN, marked at 50 mN/m. However, unlike PAN, the polar component is the primary source of surface energy in ZIF-8/PAN. This characteristic suggests that ZIF-8/PAN filters are more likely to immobilize metabolites on their surface due to their higher surface energy than CuBTC/PAN. Furthermore, the pronounced polar energy component in ZIF-8/PAN could improve immobilization efficiency for more polar analytes, offering a distinct advantage in specific applications.

### 3.3. EDS and FTIR Characterization

The elemental composition of the samples was analyzed using energy-dispersive X-ray spectroscopy (EDS). Figure 5a,c displays the electromagnetic emission spectra, quantitative analyses, and SEM images of the CuBTC/PAN and ZIF-8/PAN composites. The spectra and quantitative analyses confirm the presence of Copper and Zinc in their respective composites. Several points were selected for analysis to confirm the presence of the MOFs, where crystals or aggregates were visually evident in the fibers, where the morphology appeared as a single fiber. In both scenarios, the readings revealed the characteristic peaks of Copper and Zinc.

Figure 5b displays the FTIR spectra along the fingerprint range of the PAN, CuBTC, and CuBTC/PAN composites. The PAN electrospun fibers spectrum, in black, reveals the presence of residual DMF from the electrospinning process, evidenced by the characteristic peak of C=O at 1665 cm^−1^. The peaks ranging from 1451 to 1262 cm^−1^ are attributed to the vibrations of aliphatic C-H groups [30,32]. The pristine CuBTC, in blue, features the asymmetric stretching of the carboxylate group of the BTC ligand between 1651 and 1614 cm^−1^ and the symmetric stretching vibrations of it from 1448 to 1370 cm^−1^ [29]. The 1109 cm^−1^ peak is associated with the C-O of primary alcohols [33] and alkoxy groups C-O [34]. The 758 cm^−1^ and 728 cm^−1^ peaks correspond to the aromatic sp^2^ C-H bend vibrations [34]. According to Shaheed et al., the 1300–600 cm^−1^ peaks are linked to the out-of-plane vibrations of BTC^3+^ ions [29]. The FTIR spectrum of the CuBTC/PAN composite, shown in red, demonstrates the incorporation of CuBTC crystals within the fibers, as indicated by the pronounced peak at 1370 cm^−1^, distinctive of CuBTC, which is now discernible.

Figure 5d displays the FTIR spectra within the fingerprint region for PAN, ZIF-8, and the ZIF-8/PAN composite. The spectrum for the PAN electrospun fibers, previously described in Figure 3b, is black. The FTIR spectrum of pristine ZIF-8 exhibits the C=N stretch mode at 1583 cm^−1^, while 1444–1383 cm^−1^ peaks represent all ring stretching. The peak at 1310 cm^−1^ has been variously interpreted; Pawathi et al. associate this peak with the in-plane bending of the imidazole ring [35], while Janine et al. mention the aliphatic C-H bond in the imidazole ring [36], and Ehsan et al. explain that this peak is caused by the C-N bond [37]. The 1179, 1146, 994, and 953 cm^−1^ peaks are characterized as out-of-plane bending of the imidazole ring, indicative of C-N and C=N vibrations [36,37,38]. In red, the FTIR spectrum of the ZIF-8/PAN composite demonstrates the incorporation of ZIF-8 crystals within the fibers, as the pronounced peaks at 1179 cm^−1^ and 1146 cm^−1^, characteristic of ZIF-8, are now observable.

FTIR characterization successfully identified the characteristic peaks of the MOFs in both their pristine form and within the nanocomposite. The MOFs maintained their integral chemical features despite the potential stresses imposed by the electrospinning process, which could compromise less robust molecular structures.

### 3.4. XRD

In this study, we employ X-ray diffraction (XRD) analysis to characterize the crystalline structures of CuBTC and ZIF-8 and their respective composites produced via electrospinning. The crystal lattice of CuBTC is proven to be present on the composite as it presents the characteristic peaks of pristine CuBTC. Moreover, the preservation of the characteristic peaks is evidence that the crystal structure of the MOF was not compromised during the polymeric solution’s preparation and further electrospinning. Both samples, pristine MOF and composite, presented the typical face-centered cubic crystal lattice with typical peaks at 2θ = 11° (220), 16° (400), 17° (331), 19.5° (422), 21° (551), and 22° (440), (Figure 6a). This result is in agreement with the previously reported results [39,40]. Moreover, phase identification was assisted by the Search&Match algorithm of High Score Plus (PANanalytical Bv.) software (4.0), based on either the international Powder Diffraction File (PDF4+, Release 2015, International Centre of Diffraction Data, ICDD, Newtown Square, PA, USA) or The Cambridge Structural Database (CSD-Enterprise, version 5.37, Cambridge Crystallographic Data Centre, CCDC [29]) using the built-in powder pattern generator algorithm of the Mercury program [30]. The crystallographic parameters indicated a cubic crystal system with a space group Fm-3 m and a space group number 225, i.e., a face-centered lattice type where the lattice points are located on the corners and in the centers of all faces of the unit cell. This phase identification agrees with reports found in the literature [41,42].

Similarly, the samples containing ZIF-8 present typical well-defined peaks of crystalline ZIF-8 for the composite and the pure MOF. The XRD pattern of the ZIF-8 film for 2θ values between 5° and 40° is shown in Figure 6b. The presence of high-intensity (011), (002), (112), (022), (013), and (222) peaks verified the presence of ZIF-8 in the PAN nanofibers. These results are in agreement with preview reports [43,44]. For both MOFs, the XRD patterns evidence the crystal stability to endure the complete electrospinning process, from the stirring, mixing, relatively high temperatures, rapid solvent evaporation, and high electrostatic fields.

### 3.5. UPLC-Ms/Ms

Figure 7 shows the solid-phase extraction (SPE) methodology. The initial phase involves the preparation of human plasma samples spiked with varying concentrations of Tamoxifen (TAM) and its metabolites: 200 ng/mL of TAM, 200 ng/mL of N-desmethyltamoxifen (NDTAM), 40 ng/mL of Endoxifen (ENDO), and 20 ng/mL of 4-hydroxytamoxifen (4OHT), these values are in the average range of physiological conditions of patients under TAM treatment [45]. After sample preparation, a protein precipitation (PP) step uses three different precipitators: a ten wt.% zinc sulfate (ZnSO_4_) solution in water, acetonitrile, and methanol. These precipitators were selected based on their reported efficacy in protein precipitation rates [46]. This step ensures the denaturation of plasma proteins, specifically human serum albumin (HSA), to which up to 98% of TAM and its metabolites are bound [47]. The denaturation facilitates the direct interaction of the drugs with the electrospun composites embedded with metal-organic frameworks (MOFs) rather than with interfering biomolecules. The full method for sample pretreatment is presented in the Appendix A.

Following PP, the treated solutions undergo filtration using the specifically fabricated filters, namely, CuBTC/PAN and ZIF-8/PAN, or a control filter composed solely of PAN, housed within a reusable stainless-steel filter holder. The filtrate is then collected for subsequent analysis via ultra-performance liquid chromatography coupled with tandem mass spectrometry (UPLC-Ms/Ms). Additionally, the processes of washing and elution are conducted and preserved for analysis. The elution phase explores three different conditions: firstly, 4 mL of methanol is used due to the high solubility of the drugs in this solvent; secondly, a blend of acetonitrile:water in a 9:1 *v/v* ratio with 0.1% formic acid; thirdly, the same blend in an 8:2 *v/v* ratio. The use of acetonitrile-water mixtures mimics the binary solvent systems employed in UPLC-Ms/Ms, which can be fine-tuned to modulate the retention times of the analytes, thereby facilitating their quantification.

#### 3.5.1. Solid-Phase Extraction Optimization

We chose the ZIF-8/PAN filter as a model to simplify the optimization process and ensure resource efficiency. The ZIF-8/PAN filter thus served as a benchmark, yielding preliminary data that led to further protocol enhancements. This optimization established a standardized protocol for enhancing the reproducibility and comparability of our findings across various SPE materials.

Our study used two primary sample types: human plasma and a controlled water:methanol mixture (W:M 9:1 *v/v* with 0.1% formic acid), as shown in Figure 5. The water:methanol samples were chosen for optimization due to their simplicity, simulating an idealized environment free from plasma’s complex molecular components. In our SPE parameters’ optimization, we used 1 mL of the sample to develop a methodology that aligns with real-world clinical needs, ensuring minimally invasive sampling, resource efficiency, and the capability for serial testing in practical healthcare scenarios [48]. However, this injection volume, V_i_, was increased to 1.5 mL for plasma samples due to the addition of protein precipitators and further reconstitution.

Figure 8 illustrates the optimization of the solid-phase extraction (SPE) process using ZIF-8/PAN as an SPE cartridge using a water:methanol mixture (W:M 9:1 *v/v* with 0.1% formic acid) as an analytes carrier and methanol as an eluent. The filter mass (mg) is depicted in Figure 8a, the volumetric injection rate, Q_i_ (mL/min), in Figure 8b, the eluent volume, V_e_ (mL), in Figure 8c, and the elution volumetric rate, Q_e_ (mL/min), in Figure 8d. Each variable was tested at different levels to identify the conditions that maximize filtration and elution efficiency. The four analytes were quantified for each variable using UPLC-MS/MS, with the resulting signals integrated using the calibration curves described in Section 3.3. The SPE method developed includes a final step where the eluted solution’s volume is adjusted to the original sample volume, allowing for a direct comparison of concentrations. Retention and elution percentages are calculated using the Equations (1) and (2), respectively,
(1)$$Retention\;\%=1-\frac{C}{C_{o}}\;,$$
(2)$$Recovery\;\%=\frac{C}{C_{o}}\;,$$

*C* represents the post-process analyte concentration, and *C_o_* is the initial concentration.

Figure 8a depicts the optimization of the ZIF-8/PAN filter mass for the maximal immobilization of the analytes Tamoxifen (TAM), N-desmethyltamoxifen (NDTAM), Endoxifen (ENDO), and 4-hydroxytamoxifen (4OHTAM). The experiment spread across five trials with filter masses ranging from 1.0 to 10.0 mg, keeping the injection volume V_i_ = 1 mL and the injection volumetric flow rate Q_i_ = 0.25 mL/min fixed. The results revealed a clear optimal mass threshold for immobilization efficiency. Notably, at filter masses of 5.0 mg and above, immobilization efficiencies for TAM and NDTAM consistently exceeded 99%. The efficiency gains plateau beyond a filter mass of 5.0 mg, suggesting this is the optimal balance between material use and immobilization effectiveness. The reproducibility of results across trials highlights the ZIF-8/PAN filter’s potential for applications requiring high-efficiency compound capture. Based on these findings, subsequent experiments will utilize a filter mass of 5 mg to ensure maximum analytical performance.

Figure 8b presents TAM, NDTAM, ENDO, and 4OHT retention percentages concerning varying injection flow rates, Q_i_, while keeping the injection volume at V_i_ = 1 mL and a fixed filter mass of 0.5 mg. The graph indicates that retention remains nearly complete, around 100%, at lower flow rates of 0.25 mL/min and 0.5 mL/min. However, as flow rates rise to 1 mL/min and beyond, up to 4 mL/min, a gradual reduction in retention is observed, with percentages eventually leveling off between 80% and 90%. This suggests that while the SPE system’s retention efficiency is slightly reduced at higher flow rates, it still maintains a high level of performance. Based on these findings, a flow rate of Q_i_ = 0.5 mL/min is selected for subsequent experiments.

Figure 8c illustrates the recovery percentages of TAM, NDTAM, ENDO, and 4OHT, correlating them with the elution volume, V_e_, in milliliters while keeping the injection volume constant at V_i_ = 1 mL, as well as a filter mass of 0.5 mg and an elution volumetric flow rate Q_e_ = 0.25 mL/min. The data show that, at V_e_ values of 0.25 mL and 0.5 mL, the recovery rates for the analytes are relatively low, around 40%, indicating that elution with pure methanol is not optimally effective at these volumes. As V_e_ increases to 1 mL and 2 mL, there is a marked increase in recovery rates, achieving approximately 60% to 70%, suggesting that higher volumes enhance the elution efficiency. Notably, at V_e_ of 3 mL and 4 mL, the recovery rates peak at 100%, demonstrating that these volumes are highly effective with pure methanol, ensuring the complete recovery of the analytes. Based on these results, V_e_ = 4 mL is selected to ensure a complete recovery.

Figure 8d displays the recovery percentages for TAM, NDTAM, ENDO, and 4OHT against the elution flow rate, Q_e_, in mL/min, keeping constant the injection volume V_i_ = 1 mL, a filter mass of 0.5 mg, and an elution volumetric flow rate Q_e_ = 0.5 mL/min. The graph provides insights into the elution efficiency at various flow rates. At the lower flow rates of 0.25 mL/min and 0.5 mL/min, the recovery rates for all analytes are near-complete, approaching 100%. This suggests that the elution process is highly efficient at these flow rates, achieving a near-total recovery of the compounds. However, as Q_e_ is increased to 1 mL/min, there is a discernible decrease in recovery rates to between 85% and 75%.

Further increases in flow rate to 2 mL/min result in further declines in recovery, with rates falling between 65% and 55%. At the highest tested flow rates of 3 mL/min and 4 mL/min, the recovery rates drop below 50%, indicating a significant loss of elution efficiency. These findings suggest that elution flow rates above 0.5 mL/min are less effective, leading to a marked reduction in the recovery of the analytes.

#### 3.5.2. Protein Precipitation for Enhanced Analyte Retention and Elution

Figure 9 provides the filtration and elution efficiencies of CuBTC/PAN, ZIF-8/PAN, and PAN filters, as calculated using the percentage of retention or elution for TAM, NDTAM, ENDO, and 4OHTAM. These efficiencies are mapped against different protein precipitators, namely, ZnSO_4_ 10 wt.% in water, acetonitrile, and methanol. Additionally, a control condition employing a water:methanol (W:M) 9:1 mixture with 0.1% formic acid is included to benchmark their performance in a simulated ideal environment against a complex biological matrix.

Figure 9 indicates that the ZIF-8/PAN filter exhibits superior retention capabilities, with near-complete retention of all analytes under control conditions. On the other hand, the CuBTC/PAN filter also demonstrates relatively low retention rates for NDTAM 89.6%, ENDO 77.6%, and 4OHT 82.8%, while the PAN filter shows lower retention rates for all the analytes with values lower than 70%.

When methanol is used as a precipitator, ZIF-8/PAN filters maintain high retention rates for all analytes, around 94%, although they are slightly lower than those under controlled conditions, near 99%. Conversely, for CuBTC, the retention rates are higher for plasma samples than those of the controlled conditions for ENDO, changing from 77.6% to 86.6%, and for 4OHT, from 82.8% to 90.4%; meanwhile, for TAM and NDTAM, the retention remains practically the same. Since the control samples offer a simpler environment, this increased retention for CuBTC filters might indicate retention of the drugs in non-precipitated proteins trapped in the CuBTC surface. However, a marked decrease in retention rates is generally observed when introducing biological samples. Additionally, ZIF-8/PAN filters show higher retention for all the analytes than the CuBTC/PAN filter in control and methanol as PP samples. With ZnSO_4_ as a precipitator, the CuBTC/PAN filter exhibits the highest retention, followed by PAN and ZIF-8/PAN. Notably, this is the only scenario where PAN surpasses the retention performance of a MOF/PAN filter. However, using ZnSO_4_ as a precipitator yields substantially lower yields than methanol. Lastly, using acetonitrile as a precipitator yields the lowest analyte recovery rates in all the tests.

The effects of plasma on the SPE performance can be explained from the protein denaturing stages. In their work, Polson et al. reported that ZnSO_4_ and acetonitrile effectively denature proteins, with over 96% and 92% efficiency at a 2:1 ratio of precipitant to plasma, respectively. Although they did not provide numerical values for methanol, it is mentioned as having a similar performance to acetonitrile [46]. However, our results indicate that high precipitation yields are not the only factor important for extracting drugs from proteins. After unfolding the protein structures, the precipitant must also effectively sequester the drugs from the proteins’ lipidic structure. ZnSO_4_’s lower yields than those obtained using methanol as a precipitant can be readily explained. Due to the aqueous and ionic nature of ZnSO_4_, the non-polar and hydrophobic analytes do not naturally migrate from the proteins to the ionic liquid phase, leading to potential losses of analytes because of protein-drug affinity. Consequently, this can reduce elution yields for the analytes throughout the process.

In the cases of acetonitrile and methanol, both solvents exhibit similar precipitation yields, similar polarity, and contain organic non-polar groups in their structures. Therefore, they are expected to denature proteins and sequester drugs. However, the retention yields with acetonitrile are significantly lower than those achieved with methanol. Understanding this requires considering the processing of samples denatured with acetonitrile or methanol (as shown in Figure 7). Plasma samples typically contain about 92% water and 8% solids. After protein precipitation using a 3:1 PP:Plasma ratio, the solids are discarded, and the supernatant is retained. This solution then undergoes evaporation and is reconstituted in a water:methanol 9:1 solution to enhance analyte-composite interactions. This reconstitution occurs when the evaporated sample volume reaches 100 µL. Due to acetonitrile forming an azeotrope with water, we hypothesize that these samples will likely have a higher organic fraction, as acetonitrile is challenging to remove through evaporation [49]. After reconstitution, the final composition of the sample has a higher concentration of organic components, namely, acetonitrile and methanol. This increase in organic content will likely reduce the filter’s retention efficiency due to the strong affinity between the analytes and solvents. Conversely, samples treated with methanol are expected to be predominantly water-based in the 100 µL reconstitution step. Therefore, when these samples are filtered, the larger aqueous fraction leads to increased filter retention efficiency.

For elution, the ZIF-8/PAN filter consistently demonstrates the highest recovery rates, namely, around 98% for all the analytes in control samples and 94% in samples treated with methanol. However, using acetonitrile and ZnSO_4_ as PP drastically reduced its retention performance, with average values below 40% and 70%. The CuBTC/PAN filter shows relatively high recovery rates when ZnSO_4_ is used as the precipitating agent. As this experimental setup is designed to identify the optimal PP method for enhancing retention, a detailed analysis of the elution process is provided in the following section.

Overall, Figure 9 indicates that the best retention performance is achieved when methanol is used as the precipitating agent (PP). Under these conditions, the ZIF-8/PAN filter is a robust and efficient choice for filtration and elution, delivering high retention and recovery rates. The performance of the CuBTC/PAN filter is lower than that of the ZIF-8-based filter, and, finally, the poor performance of the PAN filter could restrict its practical applications.

#### 3.5.3. Elution Conditions for Enhanced Analytes Elution

Figure 10 illustrates the filtration yields when using methanol as PP and the elution yields of the filters for all analytes as a function of the selected eluent, either methanol or acetonitrile:water blends with different ratios (9:1 and 8:2). For all tests, samples were prepared using methanol as the protein precipitator as it demonstrated the highest retention and elution performance, as explained in the previous section of this article. Methanol is used as the reference eluent due to the analytes’ optimal solubility in it. Conversely, acetonitrile blends aim to emulate the chromatographic conditions present in UPLC, where the transition between organic and aqueous phases can be adjusted to enhance the elution of the analytes.

For the filtration step, the retention performances were consistent with those observed in Figure 9. CuBTC/PAN and ZIF-8/PAN filters showed high filtration efficiencies across all analytes (TAM, NDTAM, ENDO, 4OHT), with percentages typically above 90%. The PAN filter, however, exhibited significantly lower filtration efficiencies.

The elution step revealed more nuanced performances. When methanol was used as the elution fluid, ZIF-8/PAN demonstrated the most uniform elution rates for all the analytes with values around 90%, indicating that this solvent has a favorable affinity towards all the analytes. On the other hand, the CuBTC/PAN filter showed lower elution yields, indicating that the MOF’s features play an important role in the elution process. Moreover, the elution was not consistent along all the analytes, with elution values for TAM, NDTAM, ENDO, and 4OHT of 74.8, 88.6, 76, and 80.6%, indicating that the polarity of the analytes is not as relevant when eluting in this scenario. Due to their poor retention yields, PAN fibers’ elution rates are also poor.

We anticipated higher elution rates for ENDO and 4OHT with acetonitrile:water blends, as their higher polarity compared to TAM and NDTAM should attract them more to the blend’s aqueous phase. However, the results did not align with this prediction, and no clear trend was observed. For instance, the ZIF-8 filter more effectively eluted NDTAM with an 8:2 blend (91%) than with a 9:1 blend (62.2%), suggesting that a richer organic phase does not guarantee the elution of less polar analytes. Conversely to ZIF-8 filters, the CuBTC/PAN presented the expected results as more polar analytes, namely, ENDO and 4OHT, were preferentially eluted in acetonitrile:water blends. These findings indicate that an optimal selection of eluents and MOFs might offer a selective elution tool. Another unexpected finding was the elution of 4OHT using the CuBTC filter with 9:2 (88.6%) and 8:2 (86.4%) acetonitrile:water blends. Given 4OHT’s hydroxyl group, it was expected to elute more efficiently in blends with a higher aqueous content. However, the elution of 4OHT was practically the same with the organically richer 9:1 blends.

Although ZIF-8/PAN filters, using methanol as a precipitation agent (PP) and reconstituted in water-rich blends, as well as for elution, possess the best performance in retaining and eluting analytes from blood plasma samples, analyzing the entire system and the variables involved to explain these results is important.

The solvent effects arise from multiple non-covalent interactions between the solvent molecules and the host (MOFs) and the guest (TAM and its metabolites). In their work, Rekharsky and Inoue discuss several interactions in our systems and link the host-guest interaction to the equilibrium constant of complex formation, which relates to changes in Gibbs free energy, enthalpy, and entropy [49]. As reported in their work, for a non-polar guest interacting with a non-polar host, adsorption or host-guest interaction tends to occur spontaneously when water is the primary carrier of the analytes. This is not only due to their polarity affinity but also their repulsion towards polar water. In our system, this principle holds; after precipitation, the water-rich environment promotes immobilization within the non-polar structures of the MOFs. However, it is important to note that the MOFs’ large surface area and porosity enhance their retention capabilities compared to PAN. Although PAN is entirely non-polar, its ability to retain analytes is limited due to only weak forces acting on its flat surfaces.

In the elution process, the conditions are reversed as the aqueous content in the filters, following the washing step, is replaced with either pure organic solvents (methanol) or organic-rich binary phases (water:acetonitrile). For methanol, its small molecular size and the presence of both polar and non-polar components in its structure enable it to solvate the analytes effectively and penetrate the small pores of the MOFs, facilitating the sequestration of the analytes. This ability likely contributes to its high elution rates for all analytes. In contrast, water:acetonitrile blends interact with the analytes in a way that may initially seem counterintuitive, indicating the need for further investigation into this host-guest interaction.

So far, our discussion has primarily focused on molecular interactions affecting analyte retention and elution. However, we have to consider the macroscopic features of MOF structures. SEM characterizations revealed significant size differences between CuBTC and ZIF-8 crystals. CuBTC crystals, exhibiting a hexagonal shape, reach up to 1500 nm in their apothem, while ZIF-8 crystals do not exceed 250 nm. This size discrepancy is also mirrored in the diameter distribution of electrospun fibers, where CuBTC fibers are larger than those of ZIF-8. Consequently, although the filters have an equal mass of 5 mg, their surface area differs, likely contributing to the higher retention rates observed in ZIF-8/PAN filters.

Beyond the availability of immobilization sites, the structural rigidity of these MOFs plays a crucial role. CuBTC is known for its robust structure; however, this strength translates to greater rigidity than other MOFs [26]. Such rigidity limits the access of analyte molecules to the inner pore structures, focusing their immobilization on the MOF surface. In contrast, ZIF-8 is characterized by its well-distributed, flexible pore structures [27], enhancing its capability to immobilize molecules on the surface and within inner layers. This flexibility also aids in the release of molecules during elution steps.

Regarding the secondary building units (SBUs) of these MOFs, Figure 1 highlights that CuBTC’s SBUs are larger and more complex. This intricate assembly may lead to a higher steric effect, where spatial arrangements of atoms reduce interactions between the analytes and the CuBTC surface. Conversely, ZIF-8 has simpler, more accessible SBUs. Surface energy characterizations support this, showing that ZIF-8 possesses higher surface energy than CuBTC/PAN, similar to pure PAN.

Finally, the presence of biological entities diminishes elution yields compared to control tests. This observation suggests that some proteins may remain on the MOF surfaces, entrapping analytes and obstructing access to pores where other analytes are bound to the MOF.

## 4. Conclusions

In this study, we developed metal-organic frameworks (MOFs) based on composites with polyacrylonitrile (PAN) for the solid-phase extraction (SPE) of Tamoxifen (TAM) and its metabolites from human blood plasma. CuBTC and ZIF-8 MOFs were chosen for their robust structures and suitability for drug adsorption, while PAN provided the necessary mechanical stability. Significantly, MOF-based filters outperformed PAN filters in retention and elution, demonstrating their superior efficacy in the SPE process.

We optimized the SPE process, finding that filter masses as small as 5 mg achieved nearly 100% retention under controlled conditions and over 90% in plasma samples, although relatively high methanol volumes (4 mL) were needed for optimal recovery. The composite filters underwent a comprehensive SPE study, encompassing sample pretreatment and retention rates using methanol, acetonitrile, and ZnSO_4_ as protein-denaturing agents. Recovery rates were also analyzed using methanol and acetonitrile:water blends as eluents. Methanol protein precipitation treatment improved analyte retention. Likewise, methanol was the most effective eluent compared to other blends.

This research suggests that these MOF-based materials and the optimized SPE method could be valuable in resource-limited facilities where more complex chromatographic tools are unavailable. This opens up the possibility of implementing them in biosensing platforms for monitoring breast cancer hormonal therapy, offering personalized treatment options based on patient profiles.

## Figures and Tables

**Figure 1 nanomaterials-14-00073-f001:**
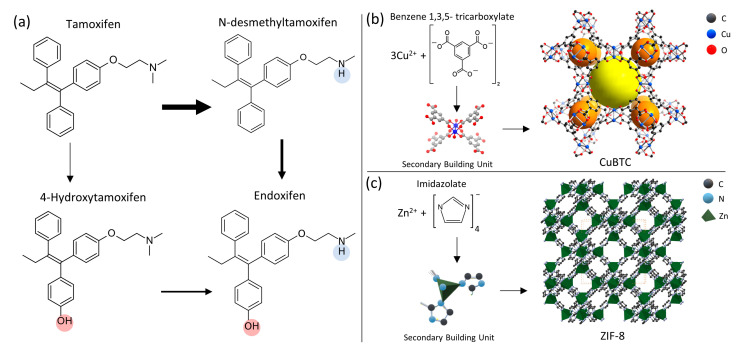
(**a**) Molecular structures of the analytes, blue and red circles helps to identify the hydroxyl and secondary amines groups between the analytes, and (**b**) CuBTC and (**c**) ZIF-8 molecular components and 3-D crystal structure, respectively.

**Figure 2 nanomaterials-14-00073-f002:**
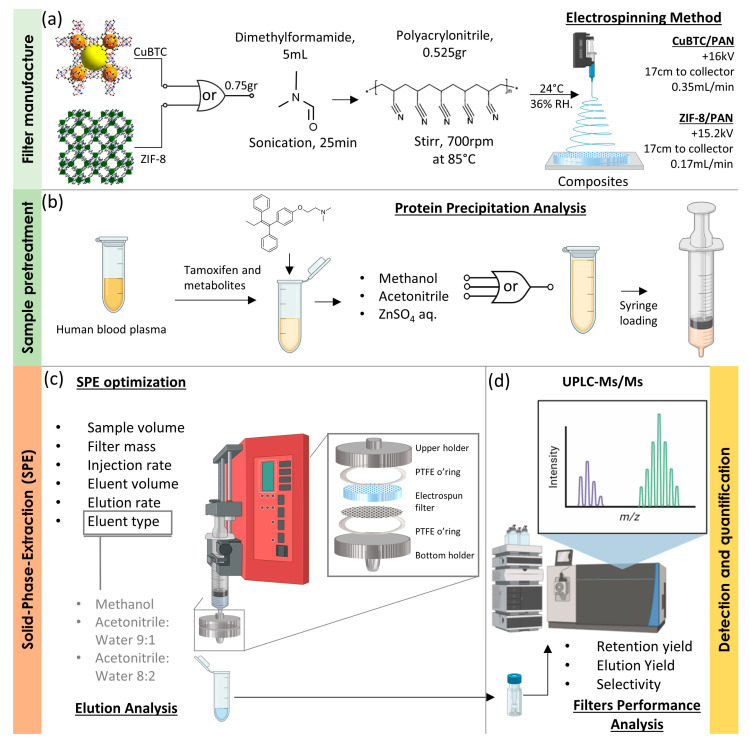
Schematic representation of the general procedure: (**a**) filter manufacture, (**b**) sample pretreatment, (**c**) solid-phase extraction optimization, and (**d**) UPLPC-Ms/Ms.

**Figure 3 nanomaterials-14-00073-f003:**
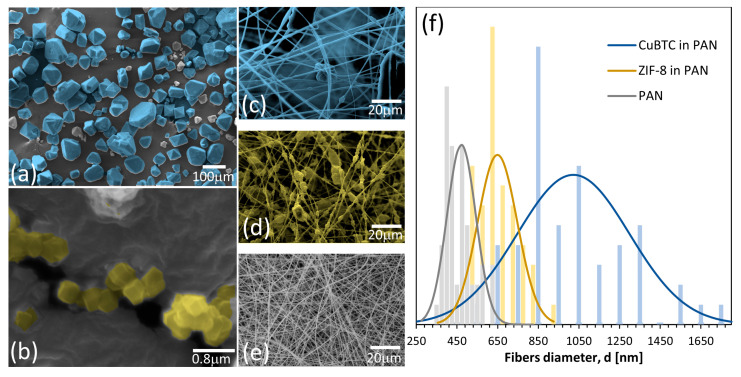
SEM images of the (**a**) pristine CuBTC, (**b**) pristine ZIF-8, (**c**) CuBTC/PAN electrospun fibers, (**d**) ZIF-8/PAN electrospun fibers, (**e**) PAN electrospun fibers, and (**f**) presents the distribution of fiber diameters of the electrospun materials.

**Figure 4 nanomaterials-14-00073-f004:**
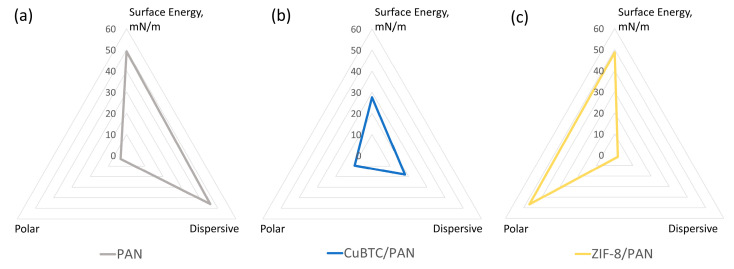
Surface energy and its polar and dispersive components of electrospun fibers of (**a**) PAN, (**b**) CuBTC/PAN, and (**c**) ZIF-8/PAN.

**Figure 5 nanomaterials-14-00073-f005:**
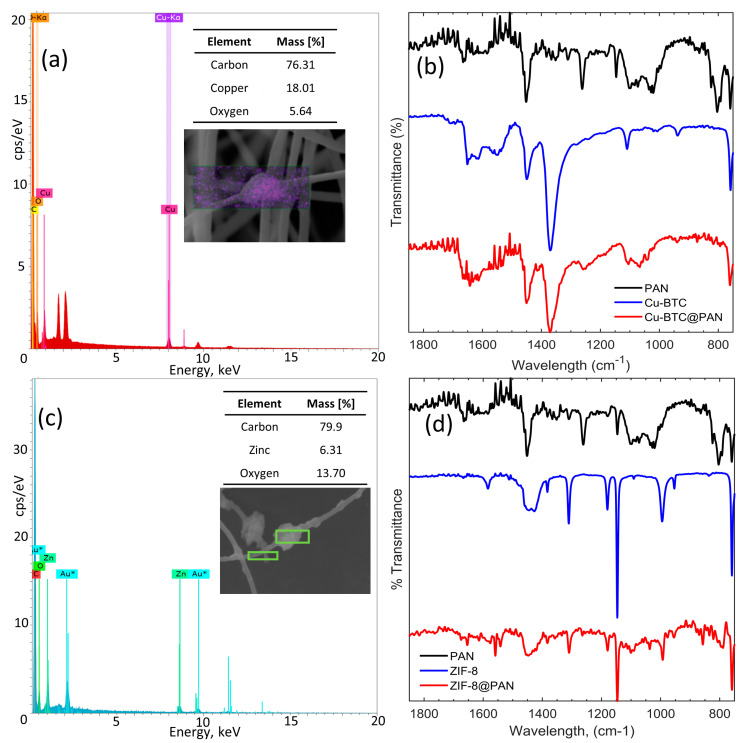
(**a**) and (**c**) are the EDS electromagnetic emission spectrum of CuBTC/PAN and ZIF-8/PAN composites. (**b**) The PAN fibers’ FTIR spectrum, pristine CuBTC, and CuBTC/PAN composite. (**d**) The FTIR spectrum of PAN fibers, pristine ZIF-8, and ZIF-8/PAN composite.

**Figure 6 nanomaterials-14-00073-f006:**
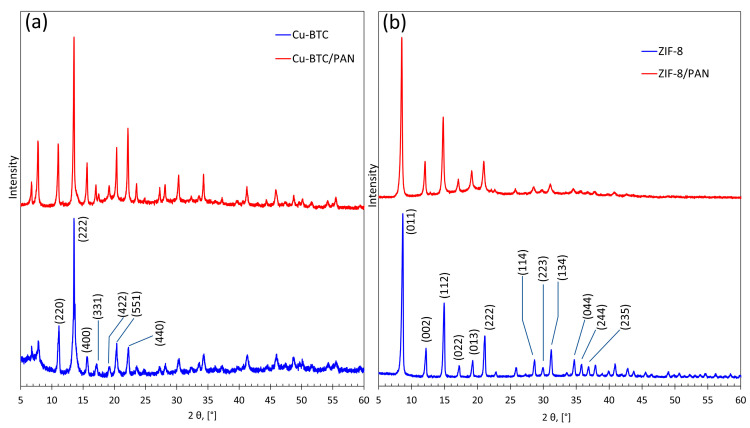
XRD patterns of (**a**) pristine CuBTC and CuBTC/PAN composites and (**b**) pristine ZIF-8 and ZIF-8/PAN composites.

**Figure 7 nanomaterials-14-00073-f007:**
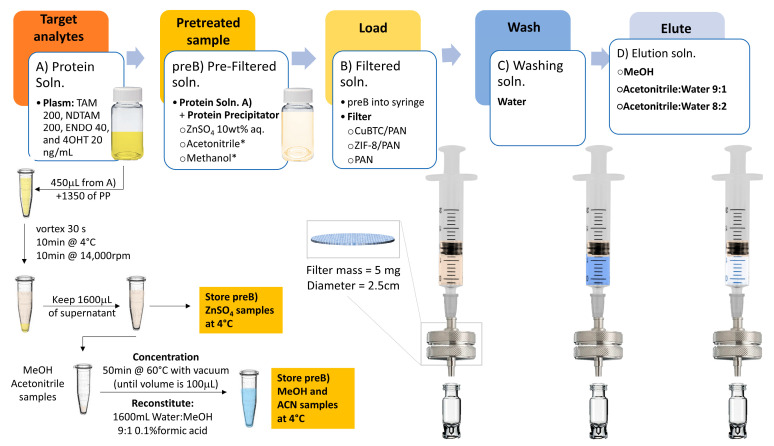
Schematic overview of the solid-phase extraction (SPE) protocol for Tamoxifen (TAM) and its metabolites from human plasma samples and sample pretreatment.

**Figure 8 nanomaterials-14-00073-f008:**
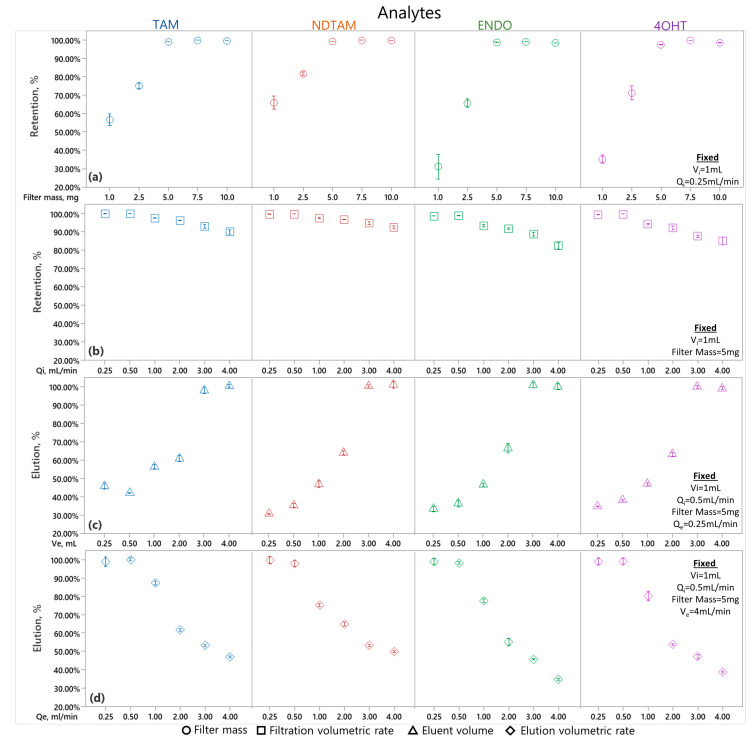
Optimization of SPE parameters using ZIF-8/PAN and water:MeOH 9:1 0.01% *v/v* formic acid as an analyte carrier and methanol as an eluent. Determination of (**a**) filter mass and (**b**) volumetric filtration on analyte retention and (**c**) eluent volume and (**d**) elution volumetric on analyte recovery.

**Figure 9 nanomaterials-14-00073-f009:**
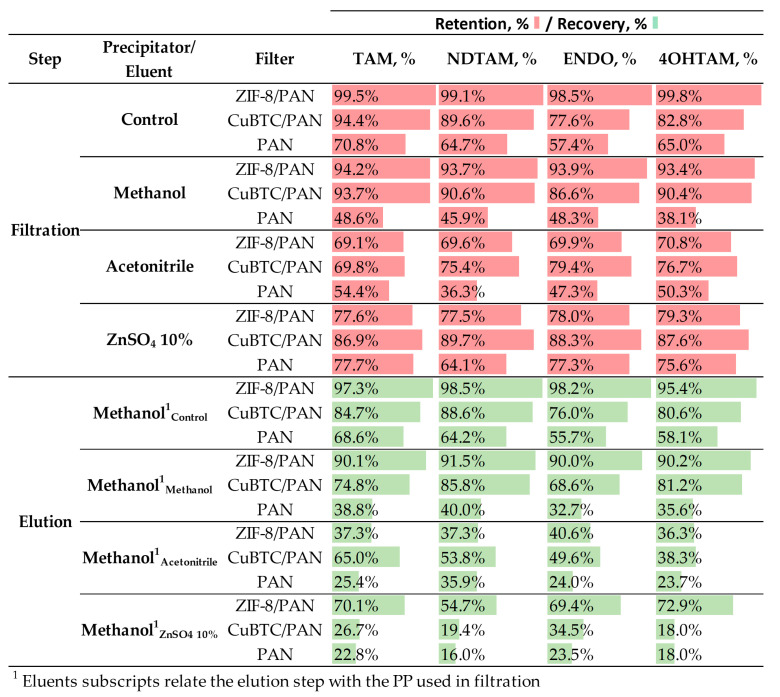
Comparative analysis of SPE filter performance using ZIF-8/PAN, CuBTC/PAN, and PAN cartridges with methanol, acetonitrile, and ZnSO_4_ as protein precipitators. SPE conditions: filter mass = 5 mg, V_i_ = 1.5 mL, Q_i_ = 0.5 mL/min, methanol V_e_ = 4 mL, and Q_e_ = 0.5 mL/min.

**Figure 10 nanomaterials-14-00073-f010:**
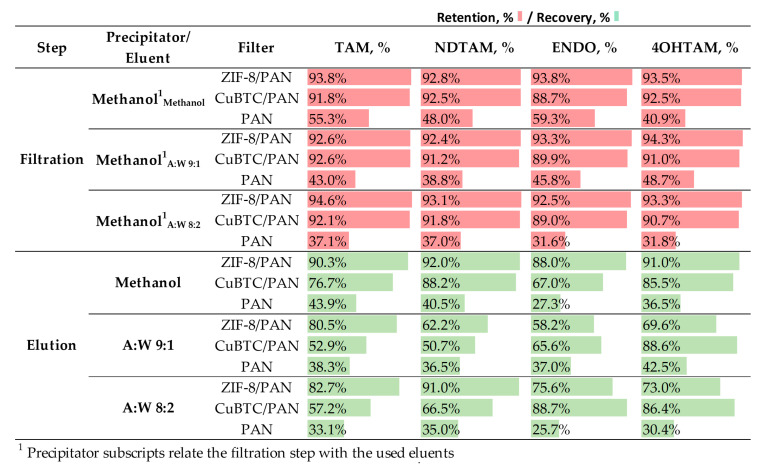
Comparative analysis of filtration and elution efficiencies using methanol and acetonitrile:water (A:W) blends as eluents. SPE conditions: filter mass = 5 mg, V_i_ = 1.5 mL, Q_i_ = 0.5 mL/min, methanol V_e_ = 4 mL, Q_e_ = 0.5 mL/min.

## Data Availability

Data is contained within the article and its Appendix A.

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
