# Peer review of "Enhancing Solid-Phase Extraction of Tamoxifen and Its Metabolites from Human Plasma Using MOF-Integrated Polyacrylonitrile Composites: A Study on CuBTC and ZIF-8 Efficacy"

_nanomaterials, 2023, doi:10.3390/nano14010073_

Round 1
Reviewer 1 Report
Comments and Suggestions for Authors
In order to extract Tamoxifen (TAM) and its three metabolites (NDTAM, ENDO, 4 OHT) more economical, convenient and efficient, two different MOFs (CuBTC and ZIF-8) materials and PAN were used to prepare the electrospun fiber composites. These composites were further synthesize the solid phase extraction (SPE) for isolation, pre-concentration, and extraction of TAM and its metabolites. Although some influence factors of SPE were discussed, and the extraction conditions were optimized, unreasonable language expression and many detailed errors severely affect the quality of the manuscript. Therefore, the manuscript needs a major modification before possible publication in nanomaterials. The comments are provided as follows:
1. In the section of “introduction”, there have no description about "solid-phase extraction (SPE)", please add a short introduction of "solid-phase extraction (SPE)" in the appropriate place to increase the readability of the manuscript.
2. The experimental results show that the properties of ZIF-8/PAN composites are better than those of CuBTC/PAN and PAN. In addition, the author also mentioned the pH-sensitive response (Page 2, line 89) of ZIF-8 material in the introduction. Why didn't study the influence of pH value on the extraction effect in the “3.4.1 Solid-Phase Extraction Optimization” part ?
3. Line 197-198, Author claimed “methanol as a non-polar solvent”, is it right?
4. Please change “ZIF8 in PAN” to “ZIF-8 in PAN” in Figure 2 (f).
5. The corresponding name is blocked by the chemical structural formula in Figure 9 (a), so please adjust it beautiful.
6. The format of “Table 1” in the article should be adjusted to center.
Comments on the Quality of English LanguageThe quality of English language needs to improve.
Author Response
Thank you very much for taking the time to review this manuscript. Please find the detailed responses in the attachment.

Reviewer 2 Report
Comments and Suggestions for Authors
My review is attached.

Minor editing of English language required
Author Response
Thank you very much for taking the time to review this manuscript. Please find the detailed responses in the attached file.

Reviewer 3 Report
Comments and Suggestions for Authors
Comments and questions
1. The authors studied the applicability of MOFs in solid-phase extraction (SPE) using MOF-integrated polyacrylonitrile nano-composites. Although it showed FTIR results, the authors did not present basic analysis results on the structure properties (crystallinity, porosity, nanostructure, etc.) of MOF and composites including MOF. Therefore, it is considered that X-ray Diffraction (XRD) results of PAN, MOF, and composite including MOF; N2 adsorption-desorption isotherm curves and pore size distributions; TEM analysis results of MOF and composite including MOF are fundamentally necessary.
2. in Figure 3: Please make the font size larger in Figure 3.
3. L239-L245: Please put a space between ‘number’ and ‘unit’.
For example) ‘1665cm-1’ → ‘1665 cm-1’
4. L244: Put ‘2’ as a superscript in ‘sp2’.
5. L246: Put ‘3+’ as a superscript in ‘BTC3+’.
6. in Figure 5: Please make the font size larger in Figure 5.
7. L345: Put a space between ‘Qi=0.5’ and ‘Qi=0.5’ in Qi=0.5mL/min.
8. L372: Put a space between ‘10’ and ‘wt.%’ in 10wt.%.
Put ‘4’ as a subscript in ‘ZnSO4’.
9. in Figure 7 and Figure 8: Please make the font size larger in Figure 7.
10. L387, L396, L607: Put ‘4’ as a subscript in ‘ZnSO4’.
11. Throughout the manuscript: Please check again for wrong grammar and typos.
-------------------------------------------------The end-------------------------------------------
Comments on the Quality of English LanguageOverall, it seems to be well-written in good-quality English. However, please check again if there are any awkward English expressions or typos.
Author Response

(The authors gave the same response as above.)

Round 2
Reviewer 1 Report
Comments and Suggestions for Authors
The authors have improved the quality of paper and answer all the questions. I suggest the paper is ready for publication.
Comments on the Quality of English LanguageQuality of English Language can be improved by jornal.
Author Response
Dear Reviewer,
Thank you for your time and valuable insights in reviewing our manuscript. We greatly appreciate your approval for publication and are grateful for your contributions to improving our work.
Reviewer 2 Report
Comments and Suggestions for Authors
The file is attached.

Extensive editing of English language required
Reviewer 3 Report
Comments and Suggestions for Authors
The authors' responses to the reviewer's questions were thought to be clear and conscientious. And it seems to have been edited to reflect those contents well in the text. Therefore, this manuscript is considered acceptable in this journal.
Author Response

(The authors gave the same response as above.)
